# Light-fueled transient supramolecular assemblies in water as fluorescence modulators

Xu-Man Chen [1], Xiao-Fang Hou[2], Hari Krishna Bisoyi [3], Wei-Jie Feng[1], Qin Cao[1], Shuai Huang[1], Hong Yang [1✉], Dongzhong Chen [2✉] & Quan Li [1,3✉]

Dissipative self-assembly, which requires a continuous supply of fuel to maintain the assembled states far from equilibrium, is the foundation of biological systems. Among a variety of fuels, light, the original fuel of natural dissipative self-assembly, is fundamentally important but remains a challenge to introduce into artificial dissipative self-assemblies. Here, we report an artificial dissipative self-assembly system that is constructed from light-induced amphiphiles. Such dissipative supramolecular assembly is easily performed using protonated sulfonato-merocyanine and chitosan based molecular and macromolecular components in water. Light irradiation induces the assembly of supramolecular nanoparticles, which spontaneously disassemble in the dark due to thermal back relaxation of the molecular switch. Owing to the presence of light-induced amphiphiles and the thermal dissociation mechanism, the lifetimes of these transient supramolecular nanoparticles are highly sensitive to temperature and light power and range from several minutes to hours. By incorporating various fluorophores into transient supramolecular nanoparticles, the processes of aggregation-induced emission and aggregation-caused quenching, along with periodic variations in fluorescent color over time, have been demonstrated. Transient supramolecular assemblies, which act as fluorescence modulators, can also function in human hepatocellular cancer cells.

[1] Institute of Advanced Materials, School of Chemistry and Chemical Engineering, and Jiangsu Province Hi-Tech Key Laboratory for Bio-medical Research, Southeast University, Nanjing, China. [2] Key Lab of High Performance Polymer Materials and Technology of MOE, School of Chemistry and Chemical Engineering, Nanjing University, Nanjing, China. [3] Advanced Materials and Liquid Crystal Institute and Chemical Physics Interdisciplinary Program, Kent State University, Kent, OH, USA. ✉email: yangh@seu.edu.cn; cdz@nju.edu.cn; quanli3273@gmail.com

In nature, active self-assemblies are widely used for various biological functions[1]. Inspired by this, scientists have an interest in utilizing knowledge of natural self-assembly mechanisms to develop artificial self-assemblies for promising applications[2]. Most artificial assemblies are constructed in a thermodynamic equilibrium state, while natural self-assemblies usually require a continuous energy supply to maintain an active and far-from-equilibrium assembled state with advanced functions. Such an assembled state is referred to as dissipative self-assembly[3,4]. According to the mechanism, a chemical reaction network is involved, including at least two irreversible chemical processes: conversion of precursors into building blocks by consuming fuel and deactivation of the building blocks to reform the precursors, accompanied by energy dissipation. Various fuels have been studied in artificial dissipative self-assemblies[5–9]; however, among all fuels, light is advantageous because it is clean, noninvasive, and remotely controlled and exhibits good spatial precision and on-demand tunability[10,11]. Azobenzenes, stilbenes, diarylethenes, and spiropyrans are usually used in photoresponsive self-assembly systems owing to the light-induced variation in steric configurations or conjugated structures upon irradiation[12,13]. Although energy-fueled amphiphilicity can successfully generate dissipative building blocks[14–16], light-induced amphiphiles have rarely been reported, especially for dissipative self-assembly.

A photoresponsive self-assembly system is an efficient approach for in situ luminescence control. To achieve photocontrollable luminescence, one method is to employ photoresponsive fluorophores as building blocks, which can lead to fluorescence variation upon light irradiation[17]. Another method is to take advantage of the Förster resonance energy transfer process[18–20]. However, these methods require covalent modification of the luminescent moieties or a good match between the fluorophores. Varying the aggregation state of fluorophores is an efficient way to modulate their luminescence while maintaining the original structures and properties of the dyes. However, it is challenging to achieve enough variation in the aggregation state to lead to obvious changes in fluorescence in light-responsive systems.

Protonated merocyanine, a photoacid, can produce a proton upon irradiation, which expands the applications of this photoresponsive system[21,22]. This property results in a tight relation between acid/base control and light control, which endows many acid/base-switchable systems with light-control properties[23–28]. On the other hand, the considerable difference in the electric dipole moment between the merocyanine and spiropyran groups provides another route for photocontrolled self-assembly[29]. Recently, merocyanine groups have been introduced into gelation systems to control the water content for photocontrolled soft actuators[30,31]. Here, we report light-fueled dissipative self-assembly based on light-induced transient amphiphiles. Inspired by the variation in the charges of the merocyanine group after irradiation, an anionic group was designed to modify protonated merocyanine to form zwitterionic protonated sulfonatomerocyanine (SMEH). SMEH can be transformed into anionic spiropyran (ASP), an anionic amphiphile, upon 420 nm irradiation, and the resulting ASP acts as an active building block for the dissipative self-assembly system. ASP spontaneously reforms SMEH in the dark (Supplementary Fig. 1). Although ASP exhibits amphiphilic properties, its amphiphilicity is not sufficient to form stable self-assembled structures in aqueous solution. Therefore, another macromolecular building block, chitosan (CS), was selected to coassemble with ASP. There are two reasons to choose CS as the other component: CS possesses many amino groups, which can bond with the protons donated by the photo-isomerized SMEH; the obtained excess ammonium can stabilize the transient assembly with ASP. Therefore, in the binary dissipative self-assembly system, SMEH isomerizes to ASP and donates a proton to CS upon 420 nm irradiation, and then positively charged CS assembles with negatively charged ASP through electrostatic and amphiphilic interactions to form transient supramolecular nanoparticles. Subsequently, the transient supramolecular nanoparticles spontaneously disassemble in the dark (Fig. 1a) owing to the relaxation of ASP to SMEH. This dissipative self-assembly system involves chemicals that are commercially available on a multigram scale and can operate in water at room temperature. Furthermore, by dynamic loading and unloading of fluorophores with aggregation-induced emission (AIE)[32] or aggregation-caused quenching (ACQ) properties, such an aqueous dissipative supramolecular self-assembly system can modulate the luminescence of dyes. Finally, an application of ASP-CS transient supramolecular nanoparticles in the staining of human hepatocellular cancer (HepG2) cells was demonstrated for dynamic cell imaging by modulating the loading of fluorophores[33,34].

## Results

**Characterization of the transient supramolecular assemblies.** First, the conversion efficiency between aqueous SMEH and ASP was investigated by UV-Vis absorption spectroscopy. As seen in Supplementary Fig. 1, the absorbance at 424 nm sharply decreased upon 420 nm irradiation for ~30 s, indicating almost complete light-induced conversion from SMEH to ASP. After keeping the abovementioned ASP solution in the dark at 25 °C, the absorbance at 424 nm gradually recovered to that of the original SMEH. According to these absorbance data as well as data reported in the literature[21], the conversion efficiency was calculated to be 95.1% (Supplementary Figs. 1 and 2), indicating an almost completely reversible conversion between the precursor SMEH and the active amphiphile ASP in aqueous solution at room temperature. Because of the isomerization of SMEH and ASP, such ASP-CS dissipative self-assembly can be performed under mild and controllable light power and temperature conditions. Then, a suitable concentration of the SMEH and CS for dissipative co-assembly was determined to be 0.15 mM SMEH/40 μg/mL CS under 420 nm irradiation (Supplementary Fig. 3). In contrast, without any of the two components or 420 nm irradiation, there was no obvious decrease in the optical transmittance at 650 nm due to the lack of amphiphilic interactions (Supplementary Fig. 4). Additionally, the 0.15 mM ASP/40 μg/mL CS solution showed an obvious Tyndall effect, which also demonstrated the formation of ASP-CS supramolecular assemblies (Supplementary Fig. 5).

The following experiments were carried out to determine the morphology and size of the ASP-CS co-assemblies. Transmission electron microscopy (TEM) images (Fig. 1d and Supplementary Fig. 6) showed several polyhedral nanoparticles of ASP-CS assemblies with an average diameter of ~170 nm upon 420 nm irradiation, agreeing well with the scanning electron microscopy (SEM) images and dynamic light scattering (DLS) results (average diameter: 179.32 nm) at 25 °C (Fig. 1b and e). However, without 420 nm irradiation, no nanoparticles were observed (Supplementary Fig. 7). The formation of polyhedral nanoparticles instead of spherical nanoparticles was mainly caused by the relatively rigid structures of ASP and CS[29]. The zeta potentials of the ASP-CS assemblies and SMEH-CS solution were −40.89 and 0.00 mV, respectively, indicating excess anions on the surface of ASP-CS assemblies but no formed assemblies in the SMEH-CS system without irradiation (Fig. 1a and c). In addition, the ASP-CS supramolecular co-assemblies could also keep their assembling states at both 15 °C and 35 °C, which showed good assembling

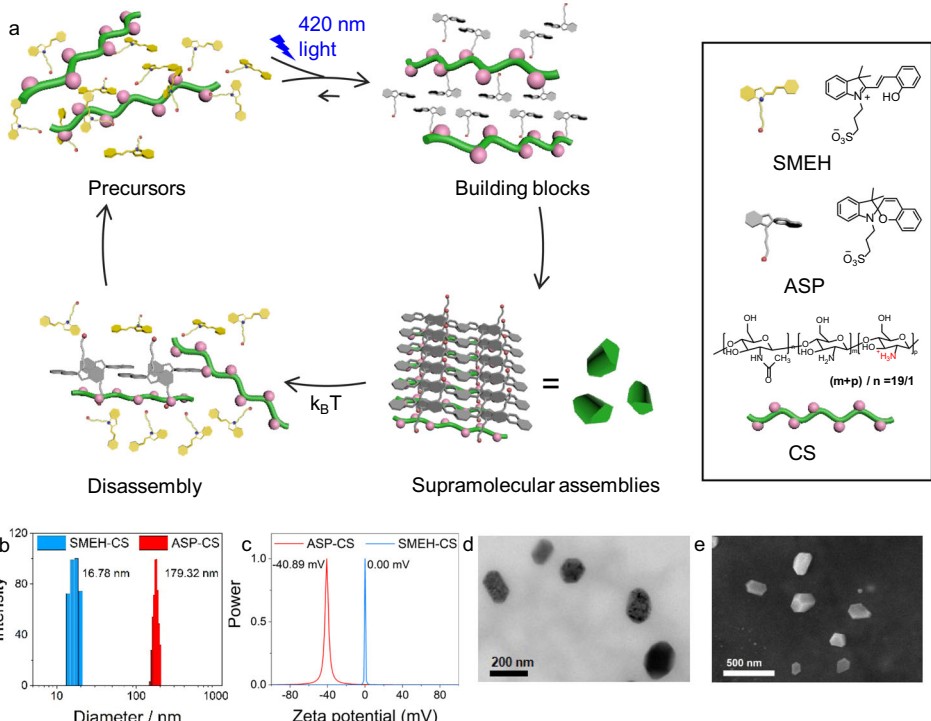

**Fig. 1 Characterization of the dissipative supramolecular self-assembly system. a** Schematic illustration of the visible-light-fueled dissipative self-assembly process. **b** Dynamic light scattering. **c** Zeta potential of SMEH-CS and ASP-CS. **d** Transmission electron microscopy. **e** Scanning electron microscopy images of ASP-CS transient nanoparticles. ([SMEH]$_{initial}$ = 0.15 mM, [CS] = 40 μg/mL).

stability towards temperature (Supplementary Fig. 8). The percentage of assembled ASP was 91.7% by measuring the UV-Vis absorption of the remaining SMEH after dialysis of ASP-CS and further recovering to SMEH in dark (Supplementary Fig. 9). Furthermore, two factors likely explain the formation of ASP-CS assemblies: the light-induced amphiphilicity of ASP and the formation of more ammonium cations of CS upon receiving protons from SMEH after irradiation. Control experiments were carried out to determine which factor was dominant. The initial pH of the SMEH-CS solution was 5.7, and the pH decreased to 4.8 after 420 nm irradiation (Supplementary Fig. 10). When we adjusted the pH of the SMEH-CS solution to 4.8 without 420 nm irradiation, the optical transmittance at 650 nm showed little decrease, but it showed an obvious decrease in ASP-CS assemblies when the pH was adjusted to 5.7 (Supplementary Fig. 11). These results suggested that light-induced amphiphilic ASP is the main component of ASP-CS co-assemblies. Electrostatic interaction between the negative charge of ASP and the positive charge of the protonated CS is the main driving force of ASP-CS co-assemblies. Besides, the hydrophobic interaction between the carbohydrate chains of CS and the spiropyran groups of ASP have also contributed for the amphiphilic co-assembly.

**Dissipative properties of the transient supramolecular assemblies**. After basic characterization of the ASP-CS assemblies induced by 420 nm irradiation, the reversible dissipative properties were determined. When the SMEH-CS solution was irradiated at 420 nm (15 mW/cm²), the optical transmittance at 650 nm decreased rapidly for ~25 s and then gradually recovered in the dark for ~1000 s at 25 °C (Supplementary Figs. 12 and 13). According to the variation in the reversible optical transmittance between the SMEH-CS solution and ASP-CS assemblies after 10 cycles, the dissipative self-assemblies exhibited good reversibility (Supplementary Fig. 14). Changes in pH are additional evidence for light-induced dissipative assembly. When the SMEH-CS

solution was irradiated with 420 nm light, the pH varied from 5.7 to 4.8, which was different from the change in the SMEH solution alone. In addition, 10 cycles of pH variation between the SMEH-CS solution and ASP-CS assemblies confirmed the good reversibility of this process (Supplementary Fig. 15). The reversible process of light-induced assembly and thermal dissociation were further investigated. As shown in Fig. 2a, the optical transmittance between SMEH-CS and ASP-CS could be repeated five times, demonstrating the stability of this dissipative process utilizing light as a fuel for transient assemblies. By comparing the transmittance of the dissipative assembly to the absorbance variation of SMEH in the SMEH-CS system, we found that the reversible process of SMEH absorbance was consistent with that of optical transmittance in the SMEH-CS dissipative system (Supplementary Fig. 16). Therefore, the absorbance variation in SMEH in the SMEH-CS system could be considered another, more distinct, way to analyze the processes in dissipative assemblies. During the process of light-induced assembly, the absorbance of SMEH-CS at 424 nm decreased slightly faster than the absorbance in a pure SMEH solution during light-induced isomerization (Supplementary Figs. 17–21). In addition, the thermal dissociation of ASP-CS (half-life: ~231 s at 25 °C) was slower than the thermal isomerization of the ASP solution alone (half-life: ~127 s at 25 °C), which indicates that the energy of ASP decreased after assembly with CS (Fig. 2b and Supplementary Fig. 21). Moreover, this dissipative self-assembly possesses a controllable lifetime that is highly sensitive to light power and temperature during formation and dissociation, respectively. Under a higher optical power density, the time required for the light-induced assembly process was obviously reduced (Fig. 2c and Supplementary Fig. 22), while the temperature had little influence on the assembly rate (Supplementary Fig. 23). On the other hand, at different temperatures, the half-life for dissociation in the dark decreased from ~908 s (15 °C) to ~54 s (35 °C) (Fig. 2d). Therefore, this light-fueled supramolecular dissipative

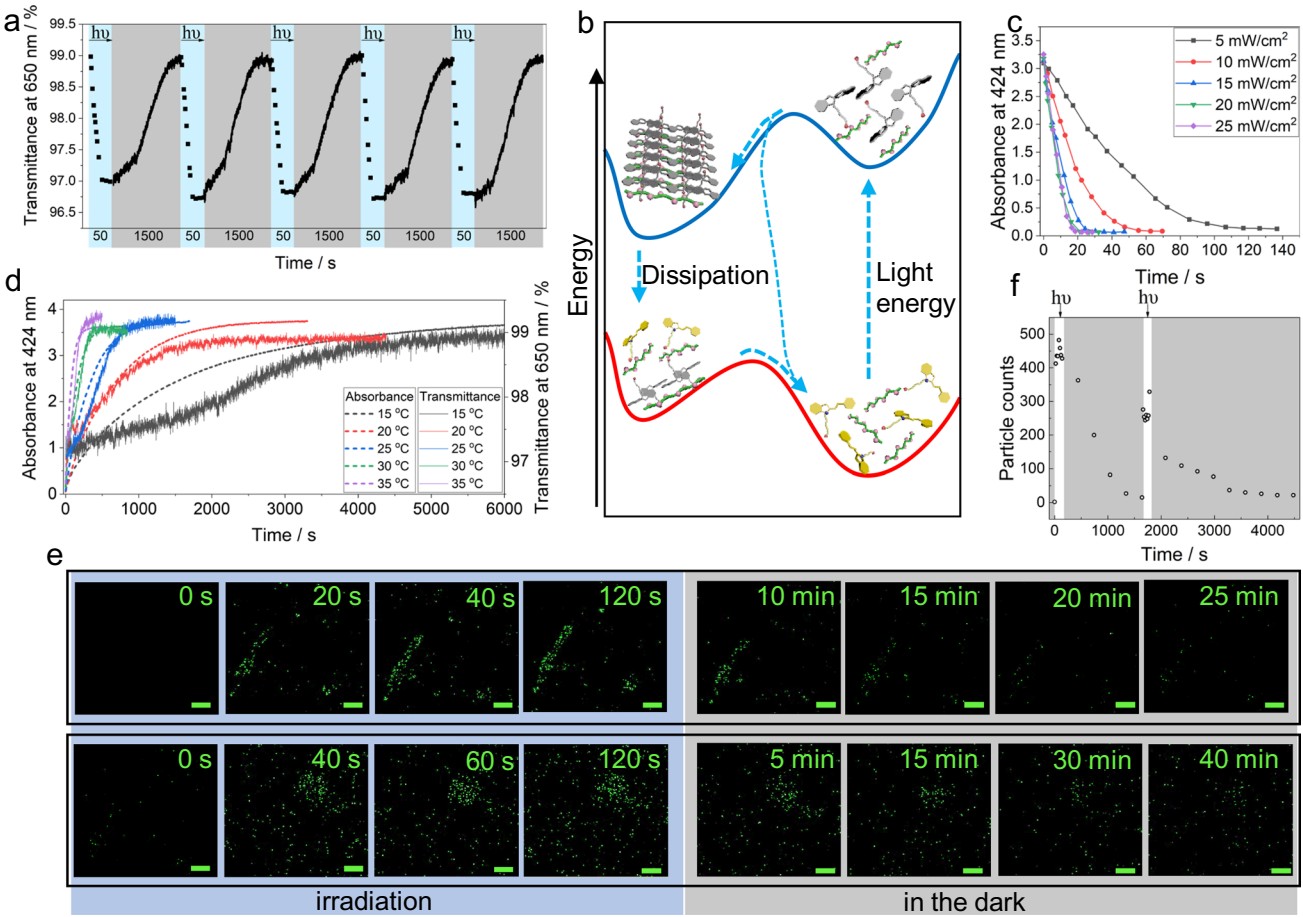

**Fig. 2 Dissipative properties of supramolecular assemblies induced by visible light. a** Optical transmittance at 650 nm for five cycles between the SMEH-CS and ASP-CS dissipative processes over time. The blue parts represent the light-induced formation of ASP-CS, and the gray parts represent the dissociation process. **b** The energy changes over the dissipative assembly process. **c** Decrease in absorbance of SMEH-CS at 424 nm upon 420 nm irradiation over time at different irradiation power densities from 5 to 25 mW/cm². **d** Variation in absorbance at 424 nm and optical transmittance at 650 nm during the thermal dissociation process of ASP-CS over time at different temperatures. **e** Partial confocal images of SMEH-CS-C153 in aqueous solution over time showing reversible formation/dissociation of fluorophore-loaded nanoparticles, including two cycles (the first cycle is on the top line, and the second cycle is on the bottom line; scale bar = 5 µm). **f** Total number of fluorescent particles detected by confocal microscopy over time (measured every 20 s for the formation process and every 5 min for the dissociation process). ([SMEH]$_{initial}$ = 0.15 mM, [CS] = 40 µg/mL, [C153] = 0.001 mM, $\lambda_{ex}$ = 405 nm, $\lambda_{em}$ = 480–520 nm, temperature: 25 °C, optical power density: 15 mW/cm² unless mentioned).

assembly system was established, including light-induced formation from an SMEH-CS solution to ASP-CS nanoparticles and thermal dissociation back to SMEH-CS, and the lifetime of this system was shown to be highly sensitive to temperature and light power.

Fortunately, the dissipative assembly process could be directly observed through laser scanning confocal microscopy (LSCM) by loading hydrophobic fluorophore coumarin 153 (C153) (Supplementary Fig. 24). Upon adding C153 (0.001 mM) to an SMEH-CS solution, the fluorescence was very weak, while bright yellow–green fluorescence was observed after 420 nm irradiation, indicating that C153 was loaded in the hydrophobic region of the ASP-CS assemblies (Supplementary Figs. 25 and 26). Then, the intensity of the induced fluorescence of C153 gradually decreased in the dark (Supplementary Fig. 27). Such periodic variation in the fluorescence of C153 was sufficiently stable for at least three cycles (Supplementary Fig. 28). Although C153 could also undergo a relative fluorescence enhancement in SMEH solution, the emission was different from that of C153 loaded in ASP-CS (Supplementary Fig. 25). Therefore, by loading C153, we could observe the emissive transient ASP-CS assemblies through LSCM. Initially, in the field of view, few distinguishable fluorescent

structures were observed, while many fluorescent points representing ASP-CS nanoparticles appeared upon 420 nm irradiation due to the accumulation of fluorescent C153 in the hydrophobic layers of ASP-CS (Fig. 2e, Supplementary Fig. 29 and Movie 1). These fluorescent nanoparticles gradually disappeared after irradiation ceased. The formation and dissociation of ASP-CS nanoparticles in two cycles were quantified by counting the number of fluorescent objects (Fig. 2f). Thus, the number of fluorescent objects rapidly increased upon irradiation and then gradually decreased.

**Time-dependent AIE and ACQ processes in the dissipative assemblies.** Moreover, light-induced dissipative assembly requires an application that can operate only in the far-from-equilibrium state. AIE and ACQ are two opposite photoluminescence mechanisms involving aggregation states. Owing to the transient aggregation state induced by light, we aimed to demonstrate time-dependent AIE and ACQ process in dissipative self-assemblies. Therefore, a typical AIE fluorophore, tetrakis(4-carboxyphenyl) ethylene (TCPE), and an ACQ fluorophore, Sulforhodamine B (SRB), were applied in the dissipative process (Fig. 3a and Supplementary Fig. 24). When TCPE (0.01 mM) was added to the

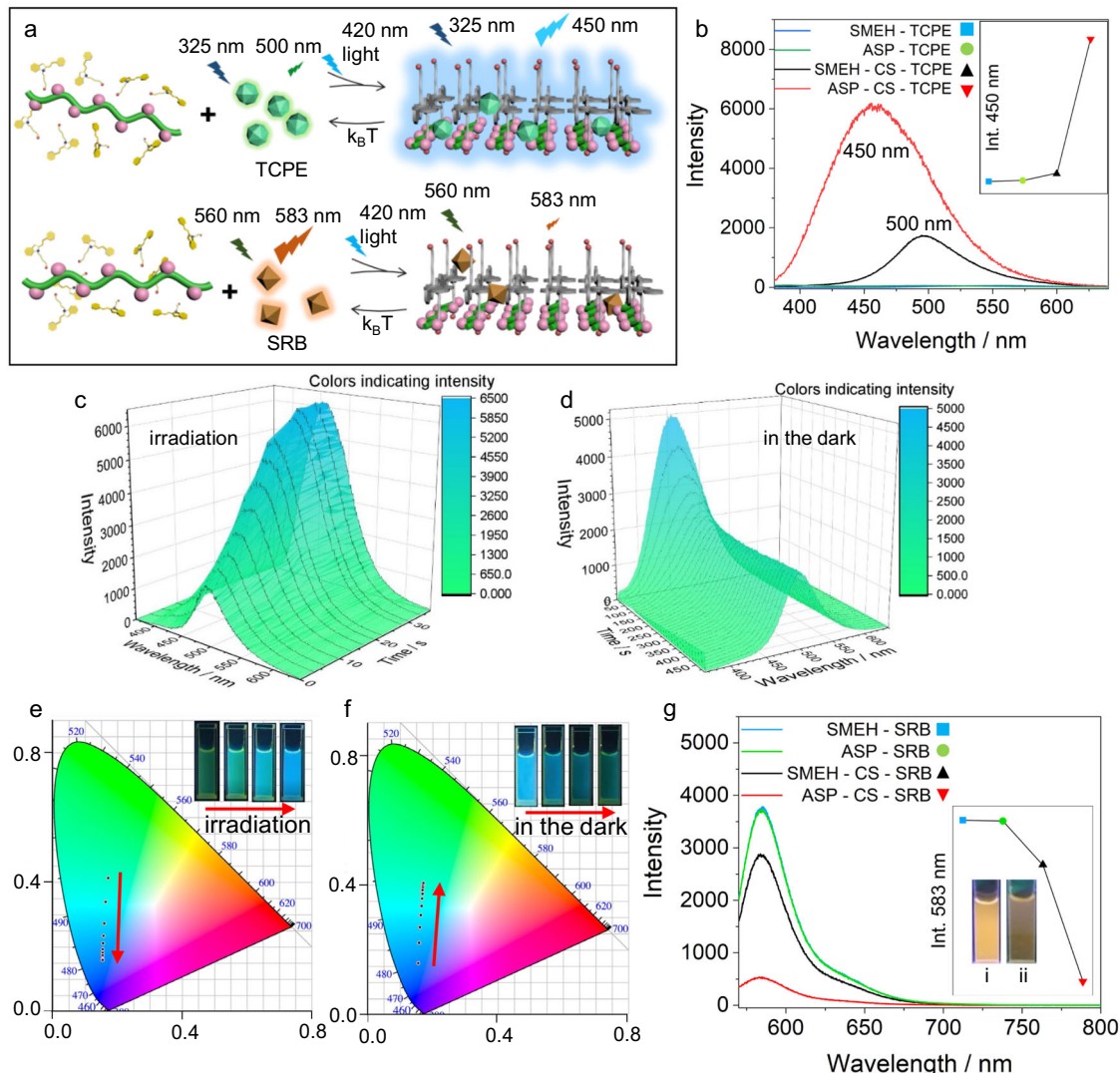

**Fig. 3 Demonstration of the time-dependent AIE and ACQ processes in dissipative self-assembly. a** Schematic illustrations of the light-induced AIE process of TCPE and the ACQ process of SRB. **b** Fluorescence emission of SMEH-TCPE, ASP-TCPE, SMEH-CS-TCPE, and ASP-CS-TCPE. Inset: emission of these solutions at 450 nm. **c** 3D fluorescence spectra of the light-induced AIE process from SMEH-CS-TCPE to ASP-CS-TCPE upon 420 nm irradiation over time. **d** 3D fluorescence spectra of the thermal dissociation process from ASP-CS-TCPE to SMEH-CS-TCPE over time. **e, f** CIE 1931 images of the light-induced AIE process and thermal dissociation process over time. Inset: fluorescence photographs of the two processes upon 365 nm UV light. **g** Fluorescence emission of SMEH-SRB, ASP-SRB, SMEH-CS-SRB, and ASP-CS-SRB. Inset: emission of these solutions at 583 nm and fluorescence photographs of (i) SMEH-CS-SRB and (ii) ASP-CS-SRB under 365 nm UV light. ($\lambda_{ex} = 325$ nm solutions containing TCPE and $\lambda_{ex} = 560$ nm solutions containing SRB unless mentioned. [SMEH]$_{initial} = 0.15$ mM, [CS] = 40 µg/mL, [TCPE] = 0.01 mM, [SRB] = 0.001 mM, temperature: 25 °C, optical power density: 15 mW/cm$^2$ unless mentioned).

SMEH-CS solution, only weak green emission (500 nm) was observed, while bright blue fluorescence (450 nm) was observed after 420 nm irradiation when TCPE was loaded into transient ASP-CS assemblies (Fig. 3b). In a control experiment, TCPE showed negligible fluorescence in SMEH solution without CS upon 420 nm irradiation. In addition, TCPE showed similar emission intensity but different emission maximum wavelength (470 nm) in only CS solution (Supplementary Fig. 30). In time-dependent fluorescence experiments with SMEH-CS-TCPE, the fluorescence at 450 nm increased quite slowly first and then increased faster and ultimately reached stable emission (Supplementary Fig. 31). When SMEH-CS-TCPE was kept in the dark after irradiation, the fluorescence of TCPE decreased much more rapidly than the dissociation of ASP-CS assemblies (Fig. 3c and d and Supplementary Fig. 32). The periodic variation in the fluorescent color over time was also indicated on the CIE 1931 image

(Fig. 3e and f). The half-life of fluorescence quenching (~35 s at 25 °C) was much shorter than that of the dissociation process, indicating that the AIE process of TCPE depends highly on the complete assembly of ASP-CS. In Supplementary Movies 2 and 3 and Fig. 3e and f, the time-dependent dynamic variation between weak green and bright blue fluorescence is shown in a cycle. When SRB (0.001 mM) was added to SMEH, ASP, CS, and SMEH-CS solutions, the emission showed little difference, while the fluorescence was obviously quenched in ASP-CS assemblies upon 420 nm irradiation (Fig. 3g and Supplementary Fig. 33). In time-dependent fluorescence experiments with SMEH-CS-SRB, the fluorescence at 583 nm decreased upon 420 nm irradiation (Supplementary Fig. 34). When SMEH-CS-SRB was kept in the dark after irradiation, the fluorescence recovery of SRB started slightly later than the dissociation of ASP-CS, indicating that the ACQ process of SRB is more sensitive to the initial aggregation of

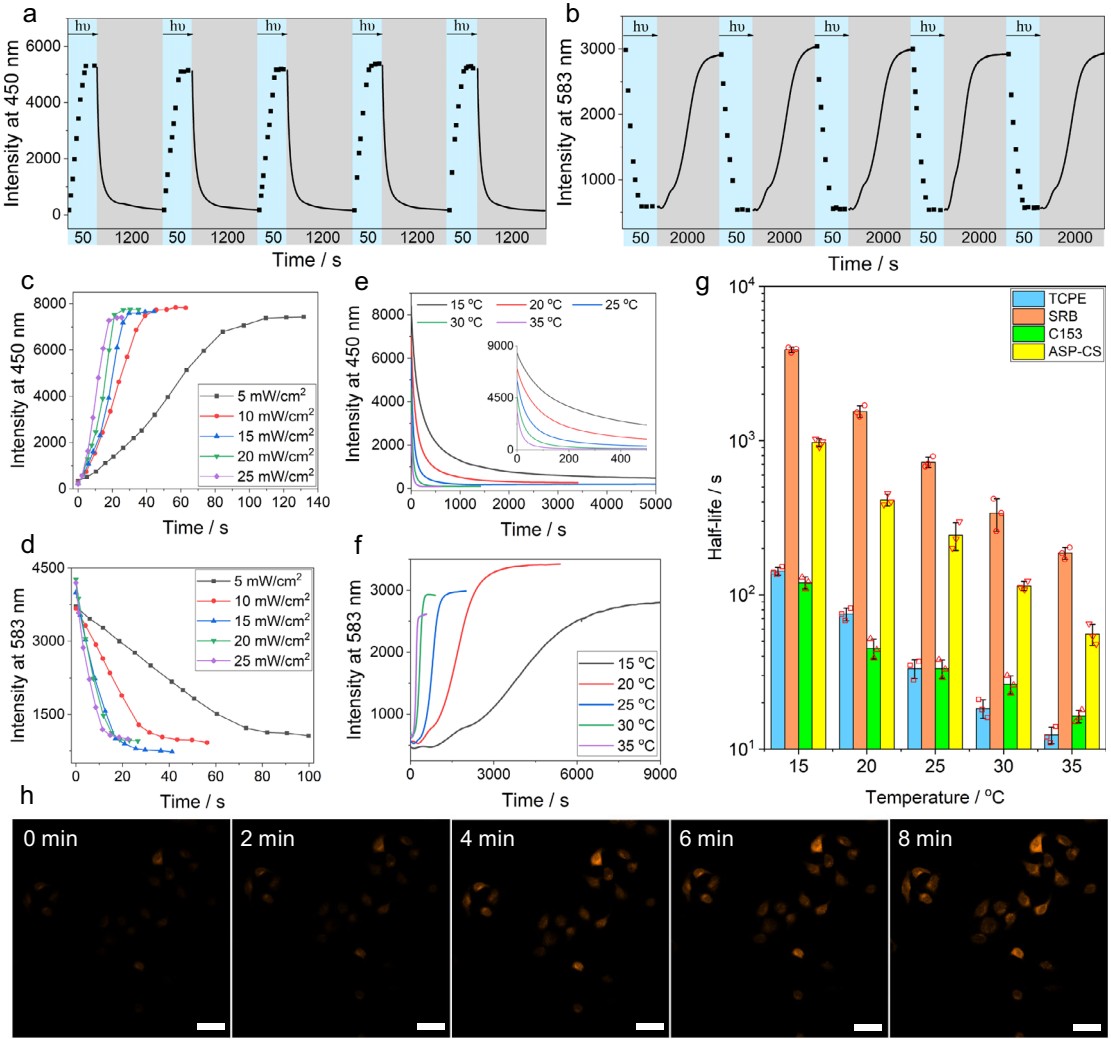

**Fig. 4 Reversible cycles, kinetics, and half-lives of fluorescence of TCPE, SRB, and C153 loaded in the dissipative assembly and dynamic cell imaging for ASP-CS-SRB. a** Reversible variation in the fluorescence of TCPE at 450 nm. **b** Reversible fluorescence variation in SRB at 583 nm for five cycles between the SMEH-CS and ASP-CS dissipative processes over time at 15 °C (optical power density: 15 mW/cm²). The blue parts represent the light-induced formation of ASP-CS, and the gray parts represent the dissociation process. **c** Variation in the fluorescence of TCPE at 450 nm. **d** Fluorescence variation in SRB at 583 nm during the formation of dissipative assembly under 420 nm irradiation with different optical power densities from 5 to 25 mW/cm² at 15 °C. **e** Variation in the fluorescence of TCPE at 450 nm. **f** Variation in the fluorescence of SRB at 583 nm during the dissociation of dissipative assembly at different temperatures from 15 °C to 35 °C. **g** Comparison of thermal dissociation half-lives at different temperatures from 15 to 35 °C detected by UV-Vis absorbance at 424 nm of ASP-CS and the fluorescence emission of TCPE (at 450 nm), SRB (at 583 nm), and C153 (at 500 nm) during thermal dissociation. $n = 3$ independent experiments, with the bar data indicating mean ± SD. ($\lambda_{ex}$(TCPE) = 325 nm, $\lambda_{ex}$(SRB) = 560 nm and $\lambda_{ex}$(C153) = 425 nm). **h** Confocal images of the dissociation process of ASP-CS-SRB in HepG2 cells at Ex. 558 nm, Em. 570–620 nm. ([SMEH]$_{initial}$ = 0.15 mM, [CS] = 40 µg/mL, [TCPE] = 0.01 mM, [SRB] = [C153] = 0.001 mM, scale bar = 20 µm).

ASP-CS (Supplementary Fig. 35). Both TCPE and SRB showed excellent fluorescent reversibility for at least five cycles due to the good reversibility of the dissipative assemblies and the photostability of the two fluorophores (Fig. 4a and b). Additionally, because the lifetime of the dissipative system could be controlled by temperature and light power, all the loaded fluorophores, namely, SRB, TCPE, and C153, exhibited controllable lifetimes of periodic fluorescence variation (Fig. 4c–f and Supplementary Figs. 36–39). The half-life of SRB during thermal dissociation ranged from 203 s (35 °C) to 3935 s (15 °C), while the half-lives of TCPE and C153 ranged from 12 to 140 s and from 17 to 105 s, respectively (Fig. 4g). Therefore, such dissipative supramolecular assemblies were able to modulate reversible light-induced time-dependent AIE and ACQ processes with different kinetics by dynamic loading of fluorophores.

**Dynamic cell imaging of the dissipative assemblies**. Furthermore, we investigated dynamic imaging of the transient assemblies in living cells. SRB was selected for loading in ASP-CS assemblies to stain human HepG2 cells. During all the staining process, the ASP-CS-SRB co-assemblies were under ~2 mW/cm² 420 nm irradiation in order to keep their assembled state. Right after staining in HepG2 cells, we put the cells under laser scanning confocal microscope and simultaneously removed the 420 nm irradiation to investigate the variation of the fluorescence imaging in the HepG2 cells. As shown in Fig. 4h, little fluorescence was observed in HepG2 cells immediately after staining with ASP-CS-SRB, while the fluorescence of SRB gradually appeared during 8 min of dissociation of the ASP-CS assemblies. Additionally, in an evaluation of the cytotoxicity of the ASP-CS-SRB assemblies through an MTT assay, the transient assemblies

showed negligible toxicity to HepG2 cells at the imaging concentration (Supplementary Fig. 40). Therefore, dynamic imaging in HepG2 cells was achieved by the fluorescence modulation of ASP-CS-SRB transient assemblies.

## Discussion

We successfully constructed a light-fueled dissipative supramolecular self-assembly system in water. SMEH was employed as a photoresponsive precursor and was further activated to induce the active building block ASP to coassemble with CS by simply mixing in an appropriate ratio. Light-induced amphiphiles have been developed as a rational strategy for the generation of transient supramolecular nanoparticles under mild and controllable conditions. Compared to previously reported dissipative assemblies fueled by chemical fuels[35], light is introduced here as a driving force to avoid producing chemical waste, and the light-fueled dissipative assembly shows more favorable self-regeneration capability. The lifetime of dissipative self-assembly is highly sensitive to temperature and light power, ranging from minutes to hours. Such periodic formation and dissociation can promote time-dependent AIE and ACQ processes by noncovalently loading TCPE and SRB, which shows good ability to control the fluorescence of various synthetic fluorophores by light through different luminescence mechanisms. By releasing SRB in HepG2 cells, dynamic imaging was achieved by the transient assemblies. These results demonstrate a possible pathway for developing and applying artificial light-fueled supramolecular dissipative assemblies as sensors, smart luminescent devices, photocatalysts, and other applications.

## Methods

**Determination of the preferable assembly concentration of ASP-CS**. The critical aggregation concentration of ASP in CS was measured by investigating the optical transmittance with keeping the concentration of CS constant (60 μg/mL) and increasing the concentration of ASP from 0 to 0.15 mM during 420 nm irradiation (Supplementary Fig. 3). We chose 650 nm wavelength to avoid the absorption of SMEH itself during the assembly and disassembly processes. The optical transmittance at 650 nm showed different linear variations with the appearance of an inflection point at 0.065 mM, which indicates that the induced critical aggregation concentration value in the presence of CS (60 μg/mL) is 0.065 mM. By varying the concentration of CS from 0 to 120 μg/mL in an ASP solution (0.15 mM), the optical transmittance at 650 nm decreased sharply before 40 μg/mL and then gradually recovered, which indicates that the preferable mixing ratio of ASP-CS is 0.15 mM ASP/40 μg/mL CS.

**Statistics and reproducibility**. Each experiment was performed with three replicates. Each measurement was taken from three distinct samples. The results indicate means ± standard deviation (SD).

**Reporting summary**. Further information on research design is available in the Nature Research Reporting Summary linked to this article.

## Data availability

The authors declare data supporting the findings of this study are available within the paper and its Supplementary Information. All data are available from the authors on reasonable request.

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

## Acknowledgements

We thank NNSFC (21971037, 22001035, 21875098, and 52003050), Jiangsu Provincial Natural Science Foundation of China (BK20190326, BK20200343), the Fundamental Research Funds for the Central Universities (2242020K40032, 2242020K40027), the Priority Academic Program Development of Jiangsu Higher Education Institutions, and "Zhishan" Scholars Programs of Southeast University for financial support.

## Author contributions

Q.L., H.Y., and D.C. directed and designed the project. X.-M.C., X.-F.H., and Q.C. performed all the experiments except for confocal microscopy studies, which were performed by W.-J.F. X.-M.C., X.-F.H., and H.K.B. analysed all the data. W.-J.F. edited all the supplementary movies. S.H. edited all the schematic illustrations. Q.L., X.-M.C., and H.K.B. co-wrote the paper. All authors discussed the results and commented on the manuscript.

## Competing interests

The authors declare no competing interests.
