## [Peer Review File · Nature Communications]

REVIEWER COMMENTS

Reviewer #1 (Remarks to the Author):

In this manuscript, Li et al. report the intriguing supramolecular dissipative self-assemblies fueled by light as fluorescent modulators. They have accomplished the light-responsive supramolecular assembly to an out-of-equilibrium form, where the elegant assembly and disassembly are observed clearly by dynamic characterization techniques including optical transmittance, fluorescence, and laser scanning confocal microscopy. It is very important that compared to previously reported dissipative assemblies fueled by chemical fuels, this light-fueled dissipative assembly system shows more favorable self-generation capability without any chemical waste. More importantly, the authors have demonstrated a new concept, i.e., employing sulfonatomerocyanine as “light-induced amphiphile” rather than the photoacid in the light-fueled dissipative assemblies. In addition, the potential application as fluorescence modulators for three fluorescent dyes with different luminescent properties has been demonstrated. The dissociation process of the transient supramolecular assemblies is also achieved in human hepatocellular cancer cells for dynamic cell imaging. Overall, this work is novel and significantly important. The manuscript is well-organized, and the experimental results are solid including various dynamic characterization. I would recommend this manuscript for publication on Nature Communications. Some minor revisions are required below:

1. The authors show that the lifetimes of dissociation process of the dissipative assembly are highly sensitive toward the temperature. As is well-known, some supramolecular assemblies are not stable at a relative higher temperature due to the non-covalent interaction. I wonder whether the ASP-CS supramolecular self-assembly can keep its assembly state at the temperature up to 35°C.
2. In Fig. 1a, the scheme is not so striking due to the small items used in the schemes, please enlarge the items for better presentation.
3. In Fig. 2e and 4h, the scale bars in each item seems the same. Please retain only one scale bar if it can represent all the sizes of the items in the figures.
4. In Supplementary Fig. 12, the two figures (a and b) should be combined for a better comparison of pH changing cycles between SMEH-CS and SMEH systems.

Reviewer #2 (Remarks to the Author):

The manuscript by Chen and coworkers described a novel dissipative assembly based on photo-induced protonation. Photo protonation using reversible photoacids has become a powerful and convenient approach to develop photo responsive assembly in recent years. This work is outstanding

comparing to many other works in several aspects. First, the photoacid coassembled with chitosan but not just in the media, which means the assembly has all the necessary functional components and a high local concentration of the photoacids. Both of them are important for using this type of assembly in different environment. Second, most of the published works on self-assembly are only for fundamental interest. This work showed that the assembly can be controlled in cell with light and can load a functional molecule, which could lead to applications in biotech or biomedical areas. Therefore, I think this work is suitable to be published in this journal after some modification and clarification as below:

1. Page 2, "The suitable concentration of the ASP and ..." ASP should be changed to SMEH since ASP was generated only after irradiation.
2. The reason for using 650 nm absorption to follow the assembly needs to be clearly explained.
3. Is it possible to determine the percentage of CS and ASP that formed assembly and the percentage of unassembled CS and ASP in the solution.
4. On the top part of page 3, pH was varied to study which of the two factors "the light-induced amphiphilicity of ASP and the formation of more ammonium cations of CS upon receiving protons from SMEH after irradiation" contributes more to the coassembly of ASP-CS. But the conclusion is "light-induced amphiphilic ASP is the main component of ASP-CS coassemblies." This conclusion is right based on the experiment result. But it does not tell which one is the main factor. Actually, it is very likely that the charge interaction between the negative charge of ASP and the positive charge of the protonated CS leads to the coassembly. The pH experiment cannot tell this because lowering pH only protonated CS but not generated negative charge on SMEH.
5. The fluorescence cell imaging is somewhat confusing. The text does not describe any irradiation. Was the system irradiated before the first image in Fig 4 was taken? (I guess so since SRB is ACQ fluorophore, but there is no description.) Did the assembly NP penetrated in cell? Or just the SRB in cell? Can fluorescence be switched off by another irradiation?

Reviewer #3 (Remarks to the Author):

In this manuscript, the authors reported an artificial dissipative self-assembly system via protonated sulfonato-merocyanine and chitosan molecular. Under illumination, the assembly can be effectively formed by electrostatic interaction and amphiphilic interaction. However, the disassembly process is spontaneous in the dark. The authors also explored the working mechanism of this artificial dissipative self-assembly system in detail. Furthermore, the application of this system in cell imaging was developed by loading dye molecules with AIE and ACQ. However, there are several areas in the manuscript that need to be further improved before this manuscript is recommended for publication in Nature Communications.

1. As a kind of polymer, chitosan molecule should not be the same size as ASP/SMEH of small molecule in the schematic diagram. Even the schematic diagram may mislead readers. It is strongly

recommended that the schematic diagram containing chitosan be adjusted accordingly (Fig.1a and Fig.3a). In addition, the deacetylation degree and molecular weight of chitosan need to be provided in the supporting information.

2. ¹H-NMR, ¹³C-NMR and HRMS of SMEH need to be provided in the supporting information.

3. The morphology and size of the ASP-CS co-assemblies is totally different in TEM images and SEM images. TEM shows spherical nanoparticles (Fig.1d), while SEM shows polyhedral particles (Fig 1e). The authors need to give a reasonable explanation, and it is better to test it again.

4. The authors mentioned in the text that the hydrophobic molecule C153 is “loaded” into the hydrophobic layer of ASP-CS assembly, and the specific operation steps need to be given (For example, what kind of solvent is used to dissolve C153).

5. The authors tested the fluorescence spectra of SMEH-TCPE and ASP-TCPE in the manuscript. However, CS as an important part of the assembly, the fluorescence spectra of CS-TCPE also need to be tested.

6. The authors described “When TCPE (0.01 mM) was added to the SMEH-CS solution, only weak green emission (500 nm) was observed, while bright blue fluorescence (450 nm) was observed after 420 nm irradiation when TCPE was loaded into transient ASP-CS assemblies.” (Page 6) The reason of blue shift of TCPE fluorescence spectrum needs to be explained.

7. SRB, TCPE and other abbreviations do not have their full names when they firstly appear.

8. It is necessary to test and explain the change of fluorescence spectrum of SRB and CS mixed system in this manuscript.

9. In the work, the artificial dissipative self-assembly system was applied to cell imaging, but does this delayed cell imaging really make sense? This may confuse readers

RE: Manuscript number: NCOMMS-21-23007

We are grateful to the three respected reviewers for their valuable time and helpful comments. We have carefully revised our manuscript and Supplementary Information by taking into account the respected reviewers' comments as appropriate.

Our point-by-point response to the three reviewers' comments and the changes made in our revised manuscript and Supplementary Information are as follows.

Reviewer #1

"In this manuscript, Li et al. report the intriguing supramolecular dissipative self-assemblies fueled by light as fluorescent modulators. They have accomplished the light-responsive supramolecular assembly to an out-of-equilibrium form, where the elegant assembly and disassembly are observed clearly by dynamic characterization techniques including optical transmittance, fluorescence, and laser scanning confocal microscopy. It is very important that compared to previously reported dissipative assemblies fueled by chemical fuels, this light-fueled dissipative assembly system shows more favorable self-generation capability without any chemical waste. More importantly, the authors have demonstrated a new concept, i.e., employing sulfonatomerocyanine as "light-induced amphiphile" rather than the photoacid in the light-fueled dissipative assemblies. In addition, the potential application as fluorescence modulators for three fluorescent dyes with different luminescent properties has been demonstrated. The dissociation process of the transient supramolecular assemblies is also achieved in human hepatocellular cancer cells for dynamic cell imaging. Overall, this work is novel and significantly important. The manuscript is well-organized, and the experimental results are solid including various dynamic characterization. I would recommend this manuscript for publication on Nature Communications. Some minor revisions are required below:"

Our response: We sincerely thank the reviewer for his/her valuable time in going through our manuscript and providing the very encouraging comments.

"The reviewer suggests some minor revisions:

1. The authors show that the lifetimes of dissociation process of the dissipative assembly are highly sensitive toward the temperature. As is well-known, some supramolecular assemblies are not stable at a relative higher temperature due to the non-covalent interaction. I wonder whether the ASP-CS supramolecular self-assembly can keep its

assembly state at the temperature up to 35 °C.”

Our response: The ASP-CS supramolecular self-assembly can keep its assembly state at the temperature up to 35 °C. We have been added the experiment and description into our revised manuscript and Supplementary Information.

“2. In Fig. 1a, the scheme is not so striking due to the small items used in the schemes, please enlarge the items for better presentation.”

Our response: Per suggestion, we have enlarged the items in Fig. 1a, 2b and 3a in our revised manuscript.

“3. In Fig. 2e and 4h, the scale bars in each item seems the same. Please retain only one scale bar if it can represent all the sizes of the items in the figures.”

Our response: We have retained only one scale bar for Fig. 2e and 4h.

“4. In Supplementary Fig. 12, the two figures (a and b) should be combined for a better comparison of pH changing cycles between SMEH-CS and SMEH systems.”

Our response: Per suggestion, we have combined the two figures together in our revised Supplementary Information.

Reviewer #2

“The manuscript by Chen and coworkers described a novel dissipative assembly based on photo-induced protonation. Photo protonation using reversible photoacids has become a powerful and convenient approach to develop photo responsive assembly in recent years. This work is outstanding comparing to many other works in several aspects. First, the photoacid coassembled with chitosan but not just in the media, which means the assembly has all the necessary functional components and a high local concentration of the photoacids. Both of them are important for using this type of assembly in different environment. Second, most of the published works on self-assembly are only for fundamental interest. This work showed that the assembly can be controlled in cell with light and can load a functional molecule, which could lead to applications in biotech or biomedical areas. Therefore, I think this work is suitable to be published in this journal after some modification and clarification as below:”

Our response: We sincerely thank the reviewer for his/her valuable time in going through our manuscript and providing the very encouraging comments.

“1. Page 2, “The suitable concentration of the ASP and ...” ASP should be changed to SMEH since ASP was generated only after irradiation.”

Our response: Per suggestion, we have changed “ASP” to “SMEH” and reorganized the corresponding description in our revised manuscript.

“2. The reason for using 650 nm absorption to follow the assembly needs to be clearly explained.”

Our response: We have given the description/explanation in our revised manuscript. The formation of nanoparticles leads to the light scattering in the solution, further inducing the decrease of optical transmittance that can be measured by UV-Vis spectrometer. So it is important to investigate the variation of transmittance at a certain wavelength which can keep away from the absorption of SMEH itself during the formation and dissociation of ASP-CS supramolecular assemblies. Thus, we have chosen 650 nm wavelength to avoid the absorption of SMEH itself during the assembly and disassembly processes.

“3. Is it possible to determine the percentage of CS and ASP that formed assembly and the percentage of unassembled CS and ASP in the solution.”

Our response: We can determine the percentage of CS and ASP that formed assembly and the percentage of unassembled CS and ASP in the solution. The corresponding statements and detailed experiment have been added into our revised manuscript and Supplementary Information.

“4. On the top part of page 3, pH was varied to study which of the two factors “the light-induced amphiphilicity of ASP and the formation of more ammonium cations of CS upon receiving protons from SMEH after irradiation” contributes more to the coassembly of ASP-CS. But the conclusion is “light-induced amphiphilic ASP is the main component of ASP-CS coassemblies.” This conclusion is right based on the experiment result. But it does not tell which one is the main factor. Actually, it is very likely that the charge interaction between the negative charge of ASP and the positive charge of the protonated CS leads to the coassembly. The pH experiment cannot tell this because lowering pH only protonated CS but not generated negative charge on SMEH.”

Our response: We agree that charge interaction between the negative charge of ASP and the positive charge of the protonated CS devotes the most for the amphiphilic co-assembly. Besides, we think the hydrophobic interaction between the carbohydrate chains of CS and the spiropyran groups of ASPs have also contributed for the amphiphilic co-assembly. Therefore, we think the driving forces of the light-induced amphiphilic co-assembly of ASP and CS can be divided into these two parts, charge interaction and hydrophobic interaction. The corresponding statements/discussion have been added into our revised manuscript.

“5. The fluorescence cell imaging is somewhat confusing. The text does not describe any irradiation. Was the system irradiated before the first image in Fig 4 was taken? (I guess so since SRB is ACQ fluorophore, but there is no description.) Did the assembly NP penetrate in cell? Or just the SRB in cell? Can fluorescence be switched off by another irradiation?”

Our response: The clearer descriptions or comments to the questions have been added into our revised manuscript and Information. The system was irradiated before the first image in Fig 4 was taken. The assembly NPs have penetrated in cell. The fluorescence cannot be switched off by another irradiation.

Reviewer #3

“In this manuscript, the authors reported an artificial dissipative self-assembly system via protonated sulfonato-merocyanine and chitosan molecular. Under illumination, *the assembly can be effectively formed* by electrostatic interaction and amphiphilic interaction. However, the disassembly process is spontaneous in the dark. The authors *also explored the working mechanism of this artificial dissipative self-assembly system in detail. Furthermore, the application of this system in cell imaging was developed by loading dye molecules* with AIE and ACQ. However, there are several areas in the manuscript that need to be further improved before this manuscript is recommended for publication in Nature Communications.”

Our response: We sincerely thank the reviewer for his/her valuable time in going through our manuscript and providing the helpful and positive comments.

“1. As a kind of polymer, chitosan molecule should not be the same size as ASP/SMEH of small molecule in the schematic diagram. Even the schematic diagram may mislead readers. It is strongly recommended that the schematic diagram containing chitosan be adjusted accordingly (Fig.1a and Fig.3a). In addition, the deacetylation degree and molecular weight of chitosan need to be provided in the supporting information.”

Our response: Per suggestion, we have enlarged the chitosan in the schematic diagrams of Fig.1a, 2b and 3a and have provided the deacetylation degree of chitosan and the average molecular weight of chitosan in our revised manuscript and Supplementary Information.

“2. *¹H-NMR, ¹³C-NMR and HRMS of SMEH need to be provided in the supporting information.*”

Our response: We have provided ¹H-NMR, ¹³C-NMR and HRMS of SMEH in our revised Supplementary Information.

“3. *The morphology and size of the ASP-CS co-assemblies is totally different in TEM images and SEM images. TEM shows spherical nanoparticles (Fig.1d), while SEM shows polyhedral particles (Fig 1e). The authors need to give a reasonable explanation, and it is better to test it again.*”

Our response: Per suggestion, we have re-measured the TEM image of ASP-CS nanoparticles and have added it into our revised manuscript. The morphology and size of the ASP-CS assemblies in TEM image is in consistence with SEM image and the dynamic light scattering data

“4. *The authors mentioned in the text that the hydrophobic molecule C153 is “loaded” into the hydrophobic layer of ASP-CS assembly, and the specific operation steps need to be given (For example, what kind of solvent is used to dissolve C153).*”

Our response: The solvent for C153 is DMSO, and the operation steps have been added into our revised Supplementary Information.

“5. *The authors tested the fluorescence spectra of SMEH-TCPE and ASP-TCPE in the manuscript. However, CS as an important part of the assembly, the fluorescence spectra of CS-TCPE also need to be tested.*”

Our response: Per suggestion, the fluorescence spectra of CS-TCPE have been tested and added into our revised Supplementary Information together with the description in our revised manuscript.

“6. *The authors described “When TCPE (0.01 mM) was added to the SMEH-CS solution, only weak green emission (500 nm) was observed, while bright blue fluorescence (450 nm) was observed after 420 nm irradiation when TCPE was loaded into transient ASP-CS assemblies.” (Page 6) The reason of blue shift of TCPE fluorescence spectrum needs to be explained.*”

Our response: The blue shift of TCPE fluorescence might result from the disappearance of the weak intermolecular interaction between SMEH and TCPE and the hydrophobic environment in the ASP-CS co-assemblies.

“7. *SRB, TCPE and other abbreviations do not have their full names when they firstly appear.*”

Our response: We have given all the full names including SRB and TCPE when they firstly appear in our revised manuscript.

“8. It is necessary to test and explain the change of fluorescence spectrum of SRB and CS mixed system in this manuscript.”

Our response: Per suggestion, we have tested the fluorescence spectra of CS-SRB solution and have added it into our revised Supplementary Information. The fluorescence of SRB showed little difference of after adding CS, indicating that there was little intermolecular interaction between SRB and CS.

“9. In the work, the artificial dissipative self-assembly system was applied to cell imaging, but does this delayed cell imaging really make sense? This may confuse readers.”

Our response: Per suggestion, the detailed description/explanation of the experiments on cell imaging has been added into our revised manuscript and Supplementary Information.

Furthermore, we have carefully checked the manuscript and the Supplementary Information. With these changes and point-by-point response to the three respected reviewers' comments, we hope that the revised manuscript is now acceptable for publication.

Your kind consideration of the revised manuscript will be greatly appreciated.

As always, thank you very much.

Stay safe and with all my best regards,

Quan

Quan Li

Institute of Advanced Materials, School of Chemistry and Chemical Engineering, and Jiangsu Province Hi-Tech Key Laboratory for Bio-medical Research

Southeast University

Nanjing 211189, China.

E-mail: quanli3273@gmail; qli1@kent.edu

and

Advanced Materials and Liquid Crystal Institute and Chemical Physics

Interdisciplinary Program

Kent State University

Kent, OH 44242, USA

<http://www.lcinet.kent.edu/users/qli180/PI/Li.htm>

REVIEWERS' COMMENTS

Reviewer #1 (Remarks to the Author):

Here, I only refer to my own previous comments. All my questions have been addressed positively by the authors.

Reviewer #2 (Remarks to the Author):

The authors have addressed all my concerns. I would like to recommend publication of this paper.

Reviewer #3 (Remarks to the Author):

The paper has been clearly improved and I would like to thank the authors for their efforts to answer all the questions raised in the first reports. In the present form, the paper is more convincing and I believe that it can be accepted as is for publication in Nature Communications.

Our point-by-point response to the three reviewers' comments is as follows.

Reviewer #1 (Remarks to the Author):

“Here, I only refer to my own previous comments. All my questions have been addressed positively by the authors.”

Our response: We greatly appreciate the reviewer for his or her valuable time and encouraging comments.

Reviewer #2 (Remarks to the Author):

“The authors have addressed all my concerns. I would like to recommend publication of this paper.”

Our response: We greatly appreciate the reviewer for his or her valuable time and encouraging comments.

Reviewer #3 (Remarks to the Author):

“The paper has been clearly improved and I would like to thank the authors for their efforts to answer all the questions raised in the first reports. In the present form, the paper is more convincing and I believe that it can be accepted as is for publication in Nature Communications.”

Our response: We greatly appreciate the reviewer for his or her valuable time and encouraging comments.

Please let us know if any further information is required.

As always, thank you so much.

Stay safe and with all my best regards,

Quan

Quan Li

Institute of Advanced Materials, School of Chemistry and Chemical Engineering, and
Jiangsu Province Hi-Tech Key Laboratory for Bio-medical Research

Southeast University

Nanjing 211189, China.

E-mail: quanli3273@gmail; qli1@kent.edu

and

Advanced Materials and Liquid Crystal Institute and Chemical Physics
Interdisciplinary Program

Kent State University

Kent, OH 44242, USA

<http://www.lcinet.kent.edu/users/qli180/PI/Li.htm>